# Predicting mosquito infection from *Plasmodium falciparum* gametocyte density and estimating the reservoir of infection

**Thomas S Churcher[1]\*, Teun Bousema[2,3], Martin Walker[1], Chris Drakeley[3], Petra Schneider[4], André Lin Ouédraogo[5], María-Gloria Basáñez[1]**

[1]Department of Infectious Disease Epidemiology, Imperial College London, London, United Kingdom; [2]Department of Medical Microbiology, Radboud University Nijmegen Medical Centre, Nijmegen, The Netherlands; [3]Department of Immunology and Infection, London School of Hygiene and Tropical Medicine, London, United Kingdom; [4]Institutes of Evolution, Immunology and Infection Research, University of Edinburgh, Edinburgh, United Kingdom; [5]Department of Biomedical Sciences, Centre National de Recherche et de Formation sur le Paludisme, Ouagadougou, Burkina Faso

**Abstract** Transmission reduction is a key component of global efforts to control and eliminate malaria; yet, it is unclear how the density of transmission stages (gametocytes) influences infection (proportion of mosquitoes infected). Human to mosquito transmission was assessed using 171 direct mosquito feeding assays conducted in Burkina Faso and Kenya. *Plasmodium falciparum* infects *Anopheles gambiae* efficiently at low densities (4% mosquitoes at 1/μl blood), although substantially more (>200/μl) are required to increase infection further. In a site in Burkina Faso, children harbour more gametocytes than adults though the non-linear relationship between gametocyte density and mosquito infection means that (per person) they only contribute slightly more to transmission. This method can be used to determine the reservoir of infection in different endemic settings. Interventions reducing gametocyte density need to be highly effective in order to halt human–mosquito transmission, although their use can be optimised by targeting those contributing the most to transmission.

\*For correspondence: thomas.churcher@imperial.ac.uk

**Competing interests:** The authors declare that no competing interests exist.

**Reviewing editor**: Prabhat Jha, University of Toronto, Canada

## Introduction

The malaria parasite is transmitted among humans by anopheline mosquitoes. Male and female transmission stages (gametocytes) are ingested by the mosquito and reproduce sexually in its stomach before developing into oocysts. Once oocysts establish, it is assumed that mosquitoes will form infectious sporozoites, so the proportion of mosquitoes developing oocysts is used as a measure of mosquito infectivity. In the human malaria parasite *Plasmodium falciparum*, mosquito infection is thought to increase with the number of gametocytes ingested by the mosquito (*Jeffery and Eyles, 1955*; *Graves et al., 1988*; *Bousema and Drakeley, 2011*). However, several studies have failed to find an association (*Boudin et al., 1993*; *Haji et al., 1996*), and the precise shape of the relationship has never been rigorously quantified, particularly at very low gametocyte densities. Difficulties arise because estimates of gametocyte density have relied on microscopy, which may miss up to 80% of parasites (*Dowling and Shute, 1966*). More sensitive molecular methods such as Pfs25mRNA quantitative nucleic acid sequence–based amplification (QT-NASBA) have been developed (*Schneider et al., 2004*). Unlike conventional microscopy, this technique enables gametocyte densities to be quantified over the entire epidemiologically relevant range.

Transmission reduction is now a key component of global efforts to control and eliminate malaria (*Alonso et al., 2011*). A wide range of novel transmission-reducing drugs and vaccines are currently

**eLife digest** Malaria is one of the world's most deadly infectious diseases. The most severe form is caused by the parasite *Plasmodium falciparum*, which can reside within red blood cells and thus evade the human immune system.

*Plasmodium* is transmitted between humans by mosquitoes. When a mosquito takes a blood meal from an individual infected with the parasite, the insect ingests *Plasmodium* gametocytes (i.e., eggs and sperm), and these go on to reproduce in the gut of the mosquito. These parasites then move to the mosquito's salivary glands, to be injected into the next person whom the mosquito bites.

Although malaria is both preventable and curable, the mortality rates in many African countries remain high, especially among children. Reducing the transmission of malaria to mosquitoes is one of the primary goals in the global effort to control and eliminate the disease. While a range of drugs and vaccines that specifically try to reduce transmission are in development, non-medical interventions such as mosquito nets and insecticide spraying can quickly and effectively reduce infection rates.

Here, Churcher et al. examine the dynamics of human to mosquito transmission of *P. falciparum*, and report that the ease with which mosquitoes become infected is not directly proportional to the density of parasite gametocytes in human blood. They found that the transmission occurs readily at very low gametocyte densities. Moreover, the transmission rate remains relatively stable as the density increases, before increasing significantly when the density reaches around 200 cells per microlitre.

Churcher et al. also challenge the assumption that children are mostly responsible for transmitting the malaria parasite by suggesting that, in certain locations, there is a more significant role for adults than previously assumed. By identifying the groups that contribute most to transmission, and targeting resources to reduce gametocyte density in those individuals, it could be possible to greatly reduce the number of infected mosquitoes and, therefore, the number of infected humans.

under development, which aim to reduce malaria incidence by restricting human to mosquito transmission. The transmission-reducing ability of front-line therapeutics (such as artemisinin-based combination therapies [ACTs], or potential combinations of ACTs with gametocytocidal drugs) is also gaining increased attention (**WHO, 2012**; **White, 2013**). It is likely that these interventions will be partially effective, but it remains unclear how much they need to reduce transmission from humans to mosquitoes in a particular location, and to which age groups they should be delivered in order to halt malaria transmission.

Mathematical models of malaria transmission tend not to include explicitly the relationship between gametocytaemia and mosquito infectivity, opting, for simplicity, to assume density-independent transmission probabilities (**Smith et al., 2007**; **Griffin et al., 2010**), or fit functions to the relationship between the number of asexual parasites and the probability of a blood-feeding mosquito becoming infected (**Ross et al., 2006**). Greater complexity, however, will be required to capture fully malaria population dynamics following the introduction of drugs and vaccines that specifically target transmission stages. For example, the World Health Organization (WHO) has recently recommended that in pre-elimination or elimination malaria programmes, single-dose primaquine (0·25 mg base per kg) with an ACT should be given to all patients with *falciparum* malaria except pregnant women and infants <1 year old (**WHO, 2012**). More detailed mathematical models will allow the full impact of this change in strategy to be assessed in a range of different endemic settings.

The optimal use of transmission-reducing interventions will also require a better understanding of the human reservoir of infection, which is likely to vary between different endemic settings. Various studies have attempted to estimate the relative contribution of different age groups to mosquito infection using skin or membrane feeding assays (**Muirhead-Thomson, 1957**; **Graves et al., 1988**; **Boudin et al., 1991**; **Githeko et al., 1992**; **Drakeley et al., 2000**; **Bonnet et al., 2003**). These studies are logistically complicated and expensive, so more efficient methods of determining the reservoir of infection are required in order to target interventions optimally. A thorough understanding of the factors determining mosquito infection would allow mathematical models to predict the relative contribution of different groups to transmission from cross-sectional surveys, both before and after the introduction of transmission-reducing interventions.

This article uses data from mosquito feeding assays conducted in Burkina Faso (**Ouédraogo et al., 2009**, Dryad: **Ouédraogo et al., 2013**) and Kenya (**Schneider et al., 2007**, Dryad: **Schneider et al., 2013**) to estimate the shape of the relationship between gametocyte density and mosquito infection

(processed data available at Dryad, *Churcher et al., 2013*). Other covariates that may influence infection such as asexual parasite density (measured by microscopy) and host age were also included to improve predictions of human to mosquito transmission. These results are combined with data from a cross-sectional survey conducted in a high transmission setting in Burkina Faso to predict the relative contribution of different age groups to overall malaria transmission.

## Results

The relationship between the number of gametocytes in the blood and mosquito infection was found to be highly non-linear (*Figure 1A*). *Plasmodium falciparum* infects mosquitoes at the very low gametocyte densities that predominate in natural infections and may not be detected using standard microscopy (*Bousema and Drakeley, 2011*). Infection rises rapidly with increasing gametocyte density, and by 1 gametocyte per microlitre, ~4% (95% Bayesian credible interval [CI], 3–5%) of all mosquitoes

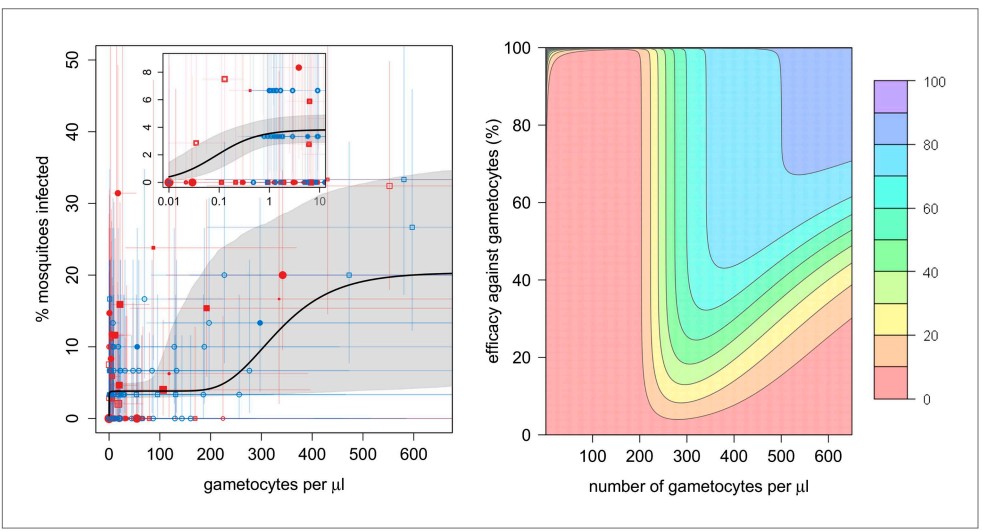

**Figure 1**. The relationship between gametocyte density and mosquito infection and the impact this will have on the effectiveness of gametocytocidal interventions. (**A**) The relationship between *Plasmodium falciparum* gametocyte density and the percentage of *Anopheles gambiae* mosquitoes that develop oocysts. Point colour, shading, and shape denote characteristics of the blood donor, such as location (blue = Burkina Faso; red = Kenya), asexual parasite density as measured by microscopy (no fill colour = none detectable, light shading = 1–1000 parasites per microlitre, dark shading ≥1000 µl⁻¹), or host age (<6 years old = square, ≥6 years old = circle). The size of the point is proportional to the number of mosquitoes dissected. Coloured horizontal and vertical lines indicate 95% Bayesian credible intervals (CIs) around point estimates. The solid black line indicates the best-fit model, whereas the grey shaded area indicates the uncertainty around this line. The inset shows the relationship at very low gametocyte densities (on a logarithmic scale). The outputs show the shape of the relationship for a child with no detectable asexual parasites. A full description of data used to fit the model are given in *Figure 1—source data 1*. Regression coefficients and a measure of goodness-of-fit of the different models are given in *Figure 1—source data 2*. Panel (**B**) shows how efficacious transmission-reducing interventions would need to be (on the vertical axis) at decreasing gametocyte density in order to reduce human to mosquito transmission (contour lines). The best-fit line from (**A**) is used to illustrate the percentage reduction in mosquito infection that would be achieved according to the pre-intervention host's gametocyte density (on the horizontal axis) for an intervention, which reduces gametocyte density by a given percentage (which is assumed to be constant over different gametocyte densities). The colours represent the percentage reduction in mosquito infection that would be achieved, ranging from red (low, 0–10% reduction) to darker hues of blue (high, 70–80% and 80–90% reduction, see legend). The 90–100% reduction in transmission is hardly visible and would correspond to nearly 100% efficacious interventions at the top of the graph.

The following source data are available for figure 1:

**Source data 1.** Description of the direct feeding assay data used to estimate mosquito infection.

**Source data 2.** Best-fit model and parameters.

develop oocysts. The subsequent increase in infection with increasing gametocyte density is best described by the Gompertz model (deviance information criterion [DIC] = 1034), which gave a significantly better fit than the linear (DIC = 1073), power (DIC = 1059), or hyperbolic (DIC = 1062) functions (*Figure 1A*). The best-fit model predicts that increasing density from 1 to 200 gametocytes per microlitre does not appreciably increase infection. Beyond 200 gametocytes per microlitre, infection rises again to finally plateau at ~18% infected mosquitoes. Gametocyte density on an arithmetic scale was a better predictor of mosquito infection than gametocyte density on a logarithmic scale. Including information on the host's asexual parasite density significantly improved model fit. Children with asexual parasite densities between 1 and 1000 parasites per microlitre were on average 27% (CI, 19–58%) less infectious to mosquitoes than those with no detectable asexual parasites, whereas those with asexual densities >1000 $\mu l^{-1}$ were 77% (CI, 14–140%) more infectious. Including age improves the fit of the model suggesting that age is an important confounder, although the Bayesian credible intervals include 0. Children >6 years old were 15% (CI, 24–64%) more likely to infect mosquitoes than those of younger ages, whereas no difference in the relationship between gametocyte density and mosquito infection was detected between Burkina Faso and Kenya.

The complex shape of the relationship between gametocyte density and mosquito infection will influence the success of transmission-reducing interventions and may explain why their ability to reduce the proportion of infected mosquitoes has been shown to depend on the gametocyte density of the host (*Churcher et al., 2012*). For example, reducing gametocyte density by 99% in a host with 200 gametocytes per microlitre may not have much effect on their immediate contribution to transmission (*Figure 1B*), although it will probably reduce the duration of infectivity. The same intervention efficacy at reducing gametocyte density in a host with 300 gametocytes per microlitre would cause an appreciable reduction in human to mosquito transmission from that individual. *Figure 1B* can be used to estimate how a reduction in the number of gametocytes will equate to a reduction in the proportion of mosquitoes becoming infected, and hence mosquito to human transmission. How this relates to the incidence of malaria and subsequent disease will depend, in part, on the degree of immunity in the human population.

In the cross-sectional survey in Burkina Faso, children harbour more gametocytes than adults, with 10 year olds having five times as many gametocytes compared to the 50 year olds (*Figure 2A*). However, the relationship between gametocyte density and mosquito infection established here means that adults are only 37% less likely to infect mosquitoes (infecting on average 3.5% of them, *Figure 2B*). Results indicate that, at the time of the survey conducted in Burkina Faso, the peak in the reservoir of infection occurs at an earlier age than the peak in gametocyte density, which will improve the (cost) effectiveness of school-based programmes.

## Discussion

The complex shape of the relationship between gametocyte density and mosquito infection elucidated here will influence our understanding of the population dynamics of *falciparum* malaria and will determine the success of transmission-reducing interventions. The efficiency with which *P. falciparum* gametocytes can infect mosquitoes means that transmission-reducing interventions, which reduce gametocyte density, will need to be highly effective in order to reduce human–mosquito transmission. Gametocytes can infect mosquitoes at very low densities, despite the mosquito needing to ingest both male and female parasites in the same blood meal. Substantial transmission was seen from hosts with <1 gametocyte per microlitre of blood. After this initial increase in infection, considerably more gametocytes are required in order to increase further the proportion of mosquitoes developing oocysts. The causes of this are unclear, but experimental systems may be informative. Data from the rodent malaria model *Plasmodium berghei* (*Sinden et al., 2007*) suggest that the phenomenon could be associated with parasite mortality during penetration of the mosquito gut wall or due to the mosquito's immune response. Beyond 200 gametocytes per microlitre mosquito infection rises again (although QT-NASBA measurement error makes the exact gametocyte density for this transition relatively uncertain). If an intervention can keep gametocyte density beneath 200 gametocytes per microlitre in a low transmission area, it may be sufficient to push the basic reproduction number of malaria beneath one and eliminate the disease. However, the highly overdispersed distribution of gametocytes between hosts (adequately described by the negative binomial distribution) means that in the population from Burkina Faso, 30% of transmission comes from hosts with densities beneath the microscopy detection threshold of 16 gametocytes per microlitre. This means that evaluating interventions

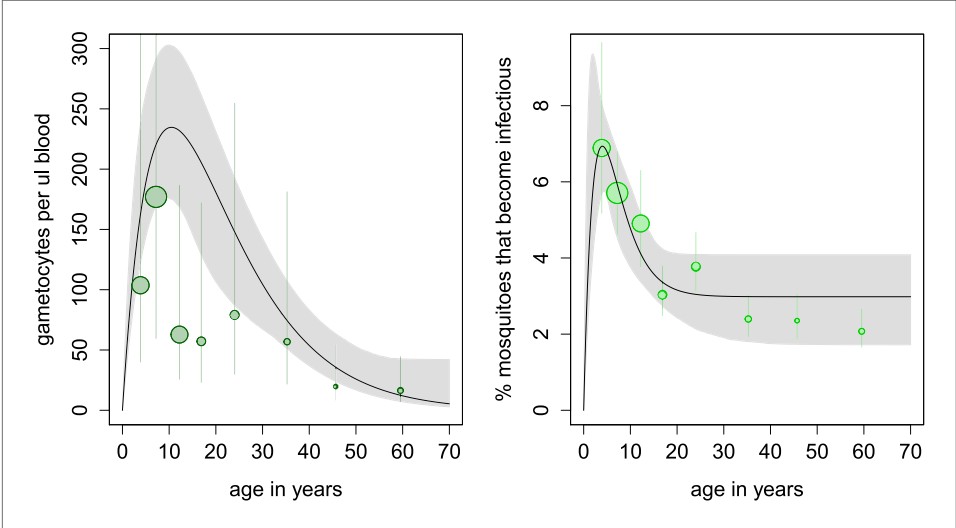

**Figure 2.** Age patterns of gametocyte density and estimates of the age profile of the human reservoir of infection. (**A**) Results from a cross-sectional survey conducted in a high transmission setting in Burkina Faso showing how the mean number of gametocytes per microlitre of blood (including 0s) changes with host age. The distribution of gametocytes among hosts is highly overdispersed. A full description of data used to fit the model are given in **Figure 2—source data 1** with the best-fit parameter estimates shown in **Figure 2—source data 2**. The relationship between gametocyte density and oocyst prevalence shown in **Figure 1A** is used to predict the percentage of mosquitoes that will become infected after biting a host of a certain age. This is shown in panel (**B**), which can be interpreted as the contribution of each age group towards the human to mosquito transmission. In both figures, the black solid line shows the best-fit model and the grey shaded area indicates the uncertainty (95% Bayesian credible Interval, CI) having fitted the model to the individual data (a total of 412 individuals). For illustrative purposes, the data are grouped into seven bins, namely 0–5, 5–10, 10–15, 15–20, 20–30, 30–40, 40–50, and ≥50 year olds, and the size of the point is proportional to the number of individuals in the group. In **Figure 1A**, the 10–15 and 15–20 year groups appear lower than the best-fit line due to sampling artefacts generated by the highly overdispersed data. Vertical lines indicate the 95% CI around grouped estimates.

The following source data are available for figure 2:

**Source data 1.** Description of the cross-sectional survey on 412 hosts carried out in Burkina Faso.

**Source data 2.** Parameter estimates for the age profile of gametocyte density and the force of infection.

solely on their ability to reduce microscopically detectable gametocytes, as has been suggested (*Graves et al., 2012*), may give misleading results.

Including information on asexual parasite density (as measured by microscopy) significantly improved the fit of the model, with hosts harbouring intermediate densities having the highest infectivity. It is unclear whether asexual parasites directly influence parasite infectivity or are associated with other (here unmeasured) variables such as the multiplicity of infection (*Nsango et al., 2012*) or factors influencing the blood environment such as immunity (*Bousema and Drakeley, 2011*). *Plasmodium falciparum* has asynchronous waves of asexual and sexual parasites. Circulating gametocytes take 2–3 days to mature before they can infect mosquitoes (*Lensen et al., 1999*), which may explain the complicated relationship between infectivity and asexual density (i.e., intermediate asexual parasite densities might occur at the start of gametocyte production and have a relatively low proportion of infectious mature gametocytes). On average, older children were more infectious, supporting previous indications that transmission-reducing immunity may predominate in young children (*Drakeley et al., 2006*).

*Figure 1A* indicates that individual QT-NASBA gametocyte density estimates are relatively uncertain. Despite this, they are considerably more accurate than conventional microscopy. Molecular techniques may be highly precise in a state-of-the-art laboratory, but the practicalities of collecting and processing samples in non-ideal field settings increases the risks of incurring measurement error.

Even though uncertainty is generally accepted by those conducting the experiments, measures of precision such as confidence intervals or standard errors around such density estimates are rarely reported in the scientific literature. Methods such as those presented in this article can allow robust quantitative insight to be gained from uncertain molecular methods. Care should be taken when interpreting gametocyte density estimates as it has been suggested that the marker used in this analysis (Pfs25 mRNA) might be parasite female specific (*Schneider, 2006*). Even if the current QT-NASBA method does only detect female gametocytes, it will neither change the qualitative conclusions of this study nor their application if other studies use the same technique. However, direct comparison of our results with those of studies using different methods for quantifying gametocyte density should be aware that the shape of the relationship may differ.

Gametocyte density measurement error is likely to cause the majority of the uncertainty seen in model outputs, although the complexity of the membrane feeding assay is also likely to contribute. The proportion of mosquitoes infected by *Plasmodium*-infected blood is known to vary substantially within and between studies. This is in part due to methodological issues with the membrane feeding assay (*Bousema et al., 2013*) but also related to biological differences in the parasite-vector combination and the blood environment (*Bousema and Drakeley, 2011*). Further standardising the membrane feeding assay, improving the accuracy of QT-NASBA technique and the inclusion of additional covariates (such as estimates of gametocyte maturity, which have been investigated in *Plasmodium vivax*; *Chansamut et al., 2012*), would improve the accuracy of the relationship between gametocyte density and mosquito infection. It would also permit the patterns described here to be checked for consistency across time and space, and enable a wider range of functional forms to be tested. Care should also be taken when interpreting the results of feeding assays as the mosquitoes used, and their biting behaviour is likely to be different from that seen in wild mosquitoes in field situations (*Bousema et al., 2013*).

Transmission reduction will become increasingly important as areas approach local elimination. Identifying host age groups that contribute most to the reservoir of infection will allow the optimal targeting of malaria control. In the high transmission site in Burkina Faso, young children harbour the majority of gametocytes but are only slightly more infective to mosquitoes than adults. The per person contribution of adults estimated here is considerably greater than that predicted by other mathematical models (*Ross et al., 2006*) and may increase further once age-dependent biting rates are taken into consideration (*Carnevale et al., 1978*; *Ross et al., 2006*). The contribution of different age groups to overall transmission will depend on local demography, age-dependent protection by malaria interventions (such as the use of bed nets or time spent in the house and therefore personally protected by indoor residual spraying), and human to mosquito contact patterns. The last two of these are difficult to measure and poorly understood, reducing our ability to accurately predict the sources of infection. The reservoir of infection is likely to vary between endemicity settings and over time so multiple cross-sectional surveys may be required. The relationship between gametocyte density and mosquito infection will allow estimates of the reservoir of infection from cross-sectional gametocytaemia surveys without the need for logistically complicated mosquito feeding assays. The results of this article should be included within mathematical models that capture changes in gametocyte density with time (*Lawpoolsri et al., 2009*) to evaluate the full impact that different transmission-reducing interventions will have on transmission and prioritise the most appropriate candidates according to transmission setting.

This article provides insights into the relationship between gametocyte density and mosquito infection that are needed to predict the outcome of transmission-reducing interventions. It shows that, given the ability of very low *P. falciparum* gametocyte densities to establish infection in *A. gambiae*, transmission-reducing interventions will need to be highly efficacious at reducing gametocyte density in order to halt human to mosquito transmission. The complex and non-linear shape of the relationship between gametocyte density and mosquito infection may explain why the ability of an interventions to reduce the proportion of infected mosquitoes has been shown to depend on the parasite load of the host (*Churcher et al., 2012*). Gametocyte density also changes with host age so the effectiveness of transmission-reducing interventions that target gametocytes will also vary with age. This should be considered in clinical trials of transmission-reducing candidate drugs and vaccines. The highly overdispersed distribution of gametocytes in the host population and the non-linear relationship between gametocytes/asexual parasite density and mosquito infection means that the impact of different transmission-reducing interventions on overall transmission at the population level will be far from intuitive.

## Material and methods

### Estimating gametocyte density and its associated uncertainty

Quantitative nucleic acid sequence-based amplification (QT-NASBA) is routinely used to estimate pathogen density. Like all diagnostic methods, it is prone to measurement error. To understand fully the associated uncertainty, it is important to appreciate the causes of the variability and how the quantification process might magnify uncertainty of point density estimates. Here, the uncertainty is quantified by repeatedly testing samples with known gametocyte density and fitting a hierarchical mathematical model.

Nucleic acids are extracted from 50 µl of blood and then amplified in the presence of a fluorescence probe. The assay measures time to positivity (TTP), which is the time it takes for the number of target amplicons detected to exceed a defined threshold (*Schneider et al., 2004*). The relationship between TTP and gametocyte density is estimated by fitting a linear regression to TTP estimates generated using a sample with known gametocyte density (a 10-fold dilution series of in vitro cultured gametocytes ranging from $10^6$ to $10^1$ gametocytes per millilitre). Let the observed TTP be denoted by $Y$ then,

$$Y = \beta_0 + \beta_1 \ln x + \varepsilon, \tag{1}$$

where $\beta_0$ and $\beta_1$ are regression coefficients estimates, $x$ is the (known) parasite density from the dilution series and $\varepsilon$ represents a normally distributed random error with mean equal to 0 and constant variance, that is $\varepsilon \sim N(0,\sigma^2)$. A full list of the parameters is given in *Table 1*. The so-called statistical calibration or inverse regression problem (*Osborne, 1991*) concerns the issue of making statistical inference on the value of an unknown (log-transformed) gametocyte density), denoted

**Table 1.** Notation of statistical and mathematical models

| Notation | Description | Equation |
|---|---|---|
| $Y$ | Time to positivity (TTP) readout generated by QT-NASBA | *Equation 1* |
| $x$ | Known density of gametocytes per millilitre (generated using dilution series) | *Equation 1* |
| $\ln \hat{x}'$ | Estimate of (unknown) gametocyte density on the logarithmic scale | *Equation 2* |
| $\hat{x}'$, | Estimate of gametocyte density on the arithmetic scale | *Equation 2* |
| $\beta_0$ | Intercept of the calibration line fitted to the dilution series | *Equation 1* |
| $\beta_1$ | Gradient of the calibration line fitted to the dilution series | *Equation 1* |
| $\sigma^2$ | Intra-assay variance measuring the accuracy with which the calibration line fits the TTP estimates from the dilution series | *Equation 1* |
| $g$ | Proportion of mosquitoes developing oocsts | *Equation 3* |
| $\varphi(\hat{x}', \kappa)$ | Saturating function determining the shape of the initial relationship between gametocytes and the proportion of mosquitoes developing oocysts | *Equation 4* |
| $f_i(\hat{x}')$ | Function determining the shape of the relationship between gametocytes and proportion of mosquitoes developing oocysts. Subscript $i$ indicates the functional form used, be it $f_\alpha(\hat{x}') = \alpha_0 + \dfrac{\alpha_1 \hat{x}'^{\alpha_2}}{(1+\alpha_3 \hat{x}'^{\alpha_2})}$, where constraining different parameters can generate a range of different shapes (constant $\alpha_1 = 0$, linear $\alpha_2 = 1$; $\alpha_3 = 0$, power $\alpha_3 = 0$, hyperbolic $\alpha_1 > 0$ $\alpha_2 = 1$ $\alpha_3 > 0$, or sigmoid $\alpha_2 > 1$) or $f_\gamma(\hat{x}') = \gamma_0 + \gamma_1 \exp\left[\gamma_2 \exp(\gamma_3 \hat{x}')\right]$, which generates a Gompertz (sigmoid-like) function | *Equation 3* |
| $\mu$ | Vector of regression coefficients | *Equation 3* |
| $z$ | Vector of dummy variables denoting donor blood characteristics, $z_1$ = asexual parasite density (0 = undetected, 1 = low, 2 = high), $z_2$ = host age (0 = younger than 6 years old, 1 = 6 or older), $z_3$ = study locale (0 = Burkina Faso, 1 = Kenya) | *Equation 3* |
| $h(A)$ | Function describing how gametocyte density and the reservoir of infection change with host age ($A$). Shape determined by parameters τ, $\psi$, and $\omega$ | *Equation 5* |

as ln $x'$, from a new TTP observation, $Y'$ The classical approach (*Eisenhart, 1939*) involves rearranging the regression model (equation 1) so that

$$\ln x' = \frac{Y' - \beta_0}{\beta_1},$$ (2)

and substituting the regression parameters with their estimates $\hat{\beta}_0$ and $\hat{\beta}_1$ to yield an estimate $\ln \hat{x}'$. Here, we adapt this approach for use in a Bayesian hierarchical model. A number of different methods have been used to do this (see *Hoadley, 1970* and *Hunter and Lamboy, 1981* and for discussion see *Osborne, 1991*), each of which has a different method for dealing with the problem of high $\hat{\beta}_1$ values (i.e., a gentle gradient, which when used in *equation 2* can generate infinitely large confidence interval estimates). Here, we use the most parsimonious approach that does not require assignment of previous distributions to (the unknown) gametocyte densities. Rather, uncertainty in the estimated regression coefficients of *equation 1* is propagated numerically via *equation 2* to yield uncertainty in the estimated gametocyte densities. Since all the calibration line data have a relatively steep gradient, the difference between the different methods will be relatively minor, and all will generate sensibly tight confidence interval estimates.

Calibration lines can vary between runs and batches of reagents. Here, we define an experiment as being a single plate run with the same batch of reagents each of which will have its own dilution series and samples with unknown density. The accuracy with which the individual TTP estimates fit the log-linear calibration line can be used to estimate assay measurement error (the intra-assay variability, $\sigma$) by fitting a hierarchical mixed-effects model to multiple dilution series (allowing estimates of $\beta_0$ and $\beta_1$ to vary among experiments to determine whether this significantly improves the fit of the model). The accuracy of gametocyte density estimates can also be improved by running multiple assays on the same sample and then taking the mean of all the ln $x'$ estimates. The application of *equation 2* yields an estimate of gametocyte density, $\hat{x}'$, on the logarithmic scale. Estimates on the arithmetic scale are generated by taking the exponent, that is $\hat{x}' = \exp(\ln \hat{x}')$. These estimates are used in the functions below to determine the relationship between gametocyte density and the proportion of mosquitoes developing oocysts.

## Mosquito infection

*Plasmodium falciparum*–infected blood was collected from children and fed through a membrane to *A. gambiae* sensu stricto mosquitoes that were dissected 7–9 days later to determine infection (oocyst carriage). Data from 171 mosquito feeds on patients' blood conducted in Burkina Faso (*Ouédraogo et al., 2009*) and Kenya (*Schneider et al., 2007*) were combined and used to fit a quantitative relationship between gametocyte density, asexual parasite density, host age, and mosquito infection (proportion of mosquitoes developing oocysts). A full description of these data is given in *Figure 1—source data 1*. Gametocyte density was quantified using QT-NASBA, and the uncertainty in the density estimates (the intra-assay variability) was quantified by fitting a hierarchical model to the 16 independent dilution series. In the Burkina Faso dataset, multiple assays were carried out on the same blood sample (an average of 2.63 assays per unknown blood sample). The mean gametocyte density from these multiple assays was taken to increase the accuracy of the estimates. The precise shape of the relationship was determined by fitting a range of different functional forms (a modified constant, linear, power, hyperbolic, sigmoid, and Gompertz functions) to data presented in *Figure 1—source data 1* and statistically determining which gave the best fit. The proportion of mosquitoes developing oocysts, denoted by $g$, can be described by *equation 3*

$$g = \varphi(\hat{x}', \kappa) f_i(\hat{x}')(1 + \mu z_1 + \mu z_2 + \mu z_3).$$ (3)

Function $f_i(\hat{x}')$ describes how infection changes with increasing estimated gametocyte density with subscript $i$ indicating the different functions tested (whose equations are given in *Table 1*); $\mu$ is a vector of regression coefficients and $z_1$, $z_2$ and $z_3$ are dummy variables denoting asexual parasite density, age of the blood donor for the membrane feeing assays, and location of provenance, respectively. Asexual parasite density (estimated by microscopy) was categorized as being either undetectable ($z_1 = 0$), low (<1000 parasites per microlitre of blood, $z_1 = 1$) or high (≥1000 parasites per microlitre of blood, $z_1 = 2$). Hosts were classified into two different age categories (<6 years [children, $z_2 = 0$] or ≥ 6 yr [older

children, $z_2 = 1$). Mosquito infection was allowed to vary between Burkina Faso and Kenya ($z_3 = 0$ for Burkina Faso, $z_3 = 1$ for Kenya). The function $\varphi(\hat{x}', \eta)$ determines the shape of the relationship at very low gametocyte densities and is motivated by the observation that the malaria parasite can adjust its sex ratio to optimise transmission (**Reece et al., 2008**). Evidence indicates that transmission is possible even at very low gametocyte densities (**Schneider et al., 2007**; **Ouédraogo et al., 2009**), so $\varphi(\hat{x}', \eta)$ allows infection to rise very quickly with increasing gametocyte density at a rate determined by parameter $\eta$,

$$\varphi(\hat{x}', \eta) = 1 - \left(1 + \frac{\hat{x}'}{2\eta}\right)^{-(\eta+1)}. \tag{4}$$

**Equation 4** was originally derived to describe the probability that a host would contain both male and female parasites according to the mean number of parasites and the aggregation (overdispersion) parameter of the negative binomial distribution (**May, 1977**). Reproduction in malaria is more complex than in the (helminth) system for which **equation 4** was devised; hence, parameter $\eta$ is not biologically interpretable.

To investigate whether mosquito infection is best predicted by gametocyte density on the arithmetic or logarithmic scale, both hypotheses were tested in the model, substituting an estimate of gametocyte density on the logarithmic scale ($\ln \hat{x}'$) for $\hat{x}'$, in **equations 3 and 4** and comparing model fits. The model quantifying the uncertainty in gametocyte density estimates was fitted at the same time as the regression determining the relationship between gametocyte density (and other covariates) and the proportion of mosquitoes infected using Bayesian Markov Chain Monte Carlo methods. Fitting the models simultaneously enables the uncertainty in the gametocyte density estimates to be reflected in the uncertainty of the shape of the relationship.

## Age profile of gametocyte density

Blood samples were randomly collected from 412 hosts from a single village in Burkina Faso (**Ouédraogo et al., 2010**). QT-NASBA assays were run on the samples, and the methods described above were used to convert assay results into estimates of gametocyte density. A description of these data is given in **Figure 2—source data 1**. To facilitate visual inspection of the age profile of gametocyte density, and to generate summary statistics, the function $h(A)$ was fitted to these data to describe how the mean number of gametocytes per microlitre of blood changes with host age ($A$)

$$h(A) = \tau + (\psi A - \tau)\exp(-\omega A). \tag{5}$$

Parameters $\tau$, $\psi$, and $\omega$ were estimated assuming a negative binomial error structure to account for the high degree of overdispersion (aggregation) observed in gametocyte density estimates (using an overdispersion parameter that did not change with host age or gametocyte density). As above, a hierarchical model was used to estimate the gametocyte density and its associated uncertainty at the same time as the age profile allowing the full uncertainty of the shape of the function to be expressed.

## Age profile of the infectious reservoir

The probability that a mosquito biting a host of a particular age will become infected can be estimated using the above model together with estimates of the host gametocyte and asexual parasite density. This was done using the cross-sectional data from Burkina Faso to generate a proxy for the human reservoir of infection at this specific time in this location (although it does not take into account how vector biting rate may vary with host age). It is important to use individual host estimates of gametocyte density instead of mean estimates as the high degree of gametocyte overdispersion among hosts may accentuate the non-linear relationship between gametocyte density and mosquito infection. For each of the 412 hosts, point estimates were obtained of the percentage of feeding mosquitoes (taking a single full blood meal) that would develop oocysts according to the age and the gametocyte/asexual parasite density of the host (**equation 3**). **Equation 5** was used to fit a relationship between host age and their contribution towards human to mosquito transmission. This best-fit line was used to predict the percentage of overall transmission that originated from hosts who had gametocyte density estimates below the detection threshold of microscopy. Here, we define the detection threshold as the minimum (positive) density that can be estimated by sampling a certain volume of blood. It is assumed

that gametocytes are counted against 500 white blood cells (WBC) and that there are on average 8000 WBC per μl of blood. This gives an average detection threshold of 16 gametocytes per microlitre of blood. This is a conservative estimate of the threshold density since densities above this value may still give false negative results both due reading errors or random sampling of gametocytes on the microscope slide.

Care should be taken when interpreting the results as mosquito membrane feeding experiments were not performed with blood from adult hosts. Evidence indicates that antibodies associated with transmission-blocking activity may be lower in older age groups (*Drakeley et al., 2006*), which could mean that adult hosts could have a greater potential to contribute to overall transmission.

## Fitting procedure

Bayesian Markov Chain Monte Carlo techniques were used to fit the models in OpenBUGS (*Lunn et al., 2009*). This approach allows the uncertainties in the assessments of gametocyte density and mosquito infection rates (generated by different numbers of mosquitoes being dissected) to be taken into consideration, propagated into parameter posterior distributions. All parameters were assigned uninformative priors and were run until convergence was reached using standard methodology (*Gelman and Rubin, 1996*). The most parsimonious models were selected by comparing deviance information criteria (DIC) values (the lowest value giving the most parsimonious yet adequate fit) (*Spiegelhalter et al., 2002*). Uncertainly around the best-fit line is indicated by showing the 95% range of 10,000 runs randomly sampled from the posterior distribution. The distribution of gametocytes between hosts was highly overdispersed, with an aggregation parameter of the negative binomial distribution of 0.185 (95% Bayesian credible interval [CI], 0.16–0.21).

# Additional information

### Funding

| Funder | Grant reference number | Author |
|---|---|---|
| European Commission FP7 Collaborative Project | HEALTH-F3-2008-223736 | Thomas S Churcher, María-Gloria Basáñez |
| AFIRM grant from the Bill & Melinda Gates Foundation | OPP1034789 | Teun Bousema, Chris Drakeley, André Lin Ouédraogo |

The funders had no role in study design, data collection and interpretation, or the decision to submit the work for publication.

### Author contributions

TSC, Conception and design, Analysis and interpretation of data, Drafting or revising the article; TB, CD, M-GB, Conception and design, Drafting or revising the article; MW, Acquisition of data, Drafting or revising the article, Contributed unpublished essential data or reagents; PS, Acquisition of data, Drafting or revising the article; ALO, Acquisition of data, Drafting or revising the article, Contributed unpublished essential data or reagents

# Additional files

### Major datasets

The following dataset was generated:

| Author(s) | Year | Dataset title | Dataset ID and/or URL | Database, license, and accessibility information |
|---|---|---|---|---|
| Churcher TS, Bousema T, Walker M, Drakeley C, Schneider P, Ouédraogo AL, Basáñez M | 2013 | Data from: Predicting mosquito infection from *Plasmodium falciparum* gametocyte density and estimating the reservoir of infection | http://dx.doi.org/10.5061/dryad.0k402 | Available at Dryad Digital Repository under a CC0 Public Domain Dedication |

The following previously published datasets were used:

| Author(s) | Year | Dataset title | Dataset ID and/or URL | Database, license, and accessibility information |
|---|---|---|---|---|
| Ouédraogo AL, Bousema T, Schneider P, de Vlas SJ, Ilboudo-Sanogo E, Cuzin-Ouattara N, Nébié I, Roeffen W, Verhave JP, Luty AJF, Sauerwein R | 2013 | Data from: Substantial contribution of submicroscopical *Plasmodium falciparum* gametocyte carriage to the infectious reservoir in an area of seasonal transmission | http://dx.doi.org/10.5061/dryad.hv01f | Available at Dryad Digital Repository under a CC0 Public Domain Dedication |
| Schneider P, Bousema JT, Gouagna LC, Otieno S, van de Vegte-Bolmer M, Omar SA, Sauerwein RW | 2013 | Data from: Submicroscopic *Plasmodium falciparum* gametocyte densities frequently result in mosquito infection | http://dx.doi.org/10.5061/dryad.589ft | Available at Dryad Digital Repository under a CC0 Public Domain Dedication |

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
