## [Decision Letter]

Thank you for sending your work entitled “Targeting Malaria Transmission Control: Predicting Mosquito Infectivity from *Plasmodium falciparum* Gametocyte Density” for consideration at *eLife*. Your article has been favorably evaluated by Prabhat Jha as Senior editor and 3 reviewers, one of whom was Vasee Moorthy.

The reviewers discussed their comments before we reached this decision and the Senior editor has assembled the following major substantive comments to help you prepare a revised submission.

**1) Estimating Gametocyte Density From QT-NASBA Assays**

I do not know the inverse regression problem well and can appreciate that there are alternative approaches. The supplementary file indicates some uncertainty in the choice of this method “for discussion see Osbourne, 1991” but this is not explained. It would be useful to give brief reasons why the particular method was chosen.

The authors say that QT-NASBA may detect only female gametocytes and elsewhere that the sex ratio of P falciparum can change to optimise transmission. This may alter the shape of the relationship between QT-NASBA results and the estimated gametocyte densities.

It would be useful to validate the model for estimating gametocyte densities from QT-NASBA assays for data other than that to what was fitted.

Figure 1 shows a double hump in the relationship between gametocytes in the blood sample and mosquito infectivity. This seems unintuitive for a biological process. I wondered to what extent this might be forced by the functions chosen. Does adding extra flexibility result in the same double hump?

Figure 1: An elegant presentation of this data. In the inset to part A there is a large clump of points on the x axis and it is not clear that the fit is good enough to be useful. For me this graph highlights the importance of understanding how different the line is in this model to other lines, and how much better this model fits, than the models that are the next best fit.

Please provide some discussion of this in the text (in addition to supplementary information). A listing of alternative models with their goodness of fit on a quantitative basis should be included in the manuscript itself.

The uncertainty range is huge, and ranges from very little increase in infectivity with increasing gametocyte density, to very steep increases in infectivity with increasing gametocyte density. What steps do the authors propose are taken to reduce the uncertainty range?

**2) Mosquito Infectivity**

Equation A4 includes a parameter called kappa. This is confusing since kappa is conventionally used to represent the infectious reservoir.

Results: The results for age are presented as if there was evidence of an effect, but the confidence intervals suggest otherwise.

The rationale for adjusting for asexual parasite density was not clear. It “improved model fit” but the aim here is not the best fitting model so much as to describe the relationship between estimated gametocyte density and infectivity. Including asexual parasite density as a main effect modifies the interpretation of the parameters.

**3) The Paper Estimates the Relationship Between Gametocyte Densities and Age**

The data comes from a cross-sectional survey of 412 people in a village in Burkina Faso. There is likely to be seasonal variation in parasite densities and the relative densities in adults and children may vary seasonally due to different levels of acquired immunity. It should be stated when the survey was carried out and also that a single survey is a potential limitation.

**4) Efficacy Definition**

A major issue is to be quite clear what is meant by efficacy throughout the paper. This lack of clarity permeates this entire technical area (i.e., not only this manuscript) and causes a lot of confusion. I believe most or all instances of use of the word efficacy in this manuscript refers to a % reduction in the gametocyte density. How does this relate to % reduction in proportion of infected mosquitoes and how will this translate to reduced incidence of human infection? This is a critical discussion that is omitted. Discussion of efficacy without some idea what this translates to in terms of reduction in human infection is rather misleading.

If the authors mean reduced transmission from humans to mosquito, I suggest they use a narrative phrase, such as this and do not use the word efficacy at all.

**5) Data/Model Description**

The data the model was fitted to was not well described (“repeatedly testing samples with known gametocyte density”). I also had trouble working out whether the fitting was simultaneous with the model for mosquito infectivity or not. The text mentions that the “estimates were used in the functions below” but the table presents all parameters together. My concern is that that the error in the estimated gametocyte densities, from the QT-NASBA results and also the chance of gametocytes being contained in the blood sample, is incorporated.

The model and parameters are not explained. The methods section merely mentions “a mathematical model” and the supplementary file gives an equation involving three parameters, but it is not clear how the equation was derived or what the parameters represent. A smooth curve is produced, but the relationship does not necessarily have to be smooth: for example, pregnant women may have higher parasitaemia and affect the age-curves.

Figure 2 suggests that the relationship does not fit well for two of the age groups. This should be acknowledged and possible reasons mentioned.

A negative binomial error structure was used. The [Supplementary-material SD3-data] table suggests that there were some zeros in the data and that the densities are skewed. Did the residuals suggest a good fit? If there are many zeros, then a zero-inflated negative binomial model may be more appropriate.

**6) Acknowledge Limitations**

On a couple of occasions, the authors make statements that are not backed up by evidence from this paper.

Discussion. It is implied that this work gives direct evidence of the effectiveness of parasite strategies at low gametocyte densities such altering the sex ratio, but no direct information on strategies is obtained.

Although the paper does include an elegant sampling approach to quantify uncertainties, Although the paper does include an elegant sampling approach to quantify uncertainties, there should be a beefed up discussion of the uncertainties underlying this work. The authors should provide more discussion on what is known about which parameters in the input data are driving uncertainty, how much confidence they have that the model they chose is superior to alternative models, and on what basis they made this decision. There should also be a discussion on the limitations related to the differences between mosquitoes used in feeding assays, and wild-type mosquitoes.

Although uncertainty is presented in one of the figure panels, and this is very helpful, uncertainty ranges/confidence intervals should be quoted in the text where figures are provided.

We suggest that the authors consider how the paper can act as an aid to conceptualising the components underlying person-to-person transmission in field settings, and to gaining a better understanding of the uncertainties related to the component of transmission that their results speak to, and also to gain a better understanding of what the existing data gaps are that must be filled in order that the research community can determine how to test new transmission-reducing interventions, and quantify their effect on person-to-person transmission.

---

## [Author Response]

***1) Estimating Gametocyte Density From QT-NASBA Assays***

*I do not know the inverse regression problem well and can appreciate that there are alternative approaches. The supplementary file indicates some uncertainty in the choice of this method “for discussion see Osbourne, 1991” but this is not explained. It would be useful to give brief reasons why the particular method was chosen*.

Inverse regression has been widely discussed in the statistical literature over the last 30 years. The majority of the controversy stems from when the gradient of the calibration line is very small (i.e., a small difference in TTP spans the whole range of gametocyte densities). This is not the case for our QT-NASBA results where the gradient is sufficiently large and no appreciable differences in the results are observed irrespective of the method used. To explain this fully to readers, we have included the following section in Supplementary file 1:

“Here we adapt this approach for use in a Bayesian hierarchical model. A number of different methods have been used to do this (see [21] and [22] and for discussion see [30]), each of which has a different approach for dealing with the problem of low beta1 values (i.e., a gentle gradient, which when used in equation A2 can generate infinitely large confidence interval estimates). Here we use the most parsimonious approach that does not require assignment of prior distributions to (the unknown) gametocyte densities. Rather, uncertainty in the estimated regression coefficients of equation A1 is propagated numerically via equation A2 to yield uncertainty in the estimated gametocyte densities. Since each of the calibration lines have a relatively steep gradient the difference between the different methods will be relatively minor and all will generate sensibly tight confidence interval estimates.”

*The authors say that QT-NASBA may detect only female gametocytes and elsewhere that the sex ratio of P falciparum can change to optimise transmission. This may alter the shape of the relationship between QT-NASBA results and the estimated gametocyte densities*.

The reviewers are correct in saying that if QT-NASBA does only detect female gametocytes then the relationship between gametocyte density (as estimated using another method which does not use Pfs25 mRNA such as microscopy) and mosquito infectivity might be different. However, we would suggest that in such a situation the number of female gametocytes would be a better prediction of infectivity than total gametocytes as each male gametocyte can fertilise up to 7 female gametocytes and the sex ratio will likely change to maximise female fertilization. Therefore, female gametocyte density represents the maximum potential infectivity.

We have changed the text accordingly:

“Even if the current QT-NASBA method does only detect female gametocytes, it will not change the qualitative conclusions of this study, nor their application if other studies use the same technique. However, direct comparison of our results with those of studies using different methods for quantifying gametocyte density should be aware that *the shape of the relationship may differ.*”

*It would be useful to validate the model for estimating gametocyte densities from QT-NASBA assays for data other than that to what was fitted*.

We are unsure of the exact purpose of this validation procedure. If the reviewers meant to validate the QT-NASBA against another highly accurate method of determining gametocyte density, then we agree that this is a necessary process for us to have full confidence in the technique and our method of correcting for its uncertainty. However, we feel that our dilution procedure (whereby a gametocyte sample is diluted 5 times and each generates a TTP) provides a strong validation procedure within the model. In this example the fit of the calibration line to these dilutions is relatively poor, which is the reason behind the uncertainty round the individual gametocyte density estimates being so high. We are in the process of refining the nucleic acid extraction and amplification technique and we are generating increasingly precise QT-NASBA estimates, as validated by the dilution procedure. Since this has been done on samples not related to the current study we feel that including these here would confuse the message. If the reviewers are suggesting that we repeat the experiment with a different dataset then we do agree, and we have highlighted this in the text. However, to our knowledge no dataset currently exists.

*Figure 1 shows a double hump in the relationship between gametocytes in the blood sample and mosquito infectivity. This seems unintuitive for a biological process. I wondered to what extent this might be forced by the functions chosen. Does adding extra flexibility result in the same double hump*?

We would argue that the double hump is consistent with our current knowledge of the biological process and what has been observed in laboratory models. The first hump is a necessity for the line to go through point 0,0 (which it has to do as zero gametocytes cannot generate any oocysts) since ∼4% of mosquitoes are infected at very low gametocyte densities. The second hump is less certain since there are relatively few data points at very high gametocyte densities. There is evidence of a second plateau in these data using differently shaped functions, which is why the power and hyperbolic functions give better fits than the linear function. This is also consistent with what has been observed in animal models (see Sinden et al. 2009). That said, we agree that the functions chosen might influence the shape and that it could be more complicated than shown here.

More complex functions were tested (such as asymmetrical logistic functions), though they failed to converge adequately due to the additional parameters and the high uncertainty in gametocyte density. We have added a comment to stress that a wider range of functions should be tested when the accuracy of the data allows:

Discussion: “Further standardising the membrane feeding assay, improving the accuracy of QT-NASBA technique and the inclusion of additional covariates (such as estimates of gametocyte maturity which have been investigated in *P. vivax* (8)) would improve the accuracy of the relationship between gametocyte density and mosquito infectivity. It would permit to see if the patterns described here are consistent across time and space, and enable a wider range of functional forms to be tested.”

*Figure 1: An elegant presentation of this data. In the inset to part A there is a large clump of points on the x axis and it is not clear that the fit is good enough to be useful. For me this graph highlights the importance of understanding how different the line is in this model to other lines, and how much better this model fits, than the models that are the next best fit*.

*Please provide some discussion of this in the text (in addition to supplementary information). A listing of alternative models with their goodness of fit on a quantitative basis should be included in the manuscript itself*.

We agree that this information should be presented in the main text and we have added the following sentences to the Results section:

“Infectivity rises rapidly with increasing gametocyte density and by 1 gametocyte µl-1 ∼4% of all mosquitoes develop oocysts. The subsequent increase in infectivity with increasing gametocyte density is best described by the Gompertz model (deviance information criterion, DIC = 1034), which gave a significantly better fit than the linear (DIC=1073), power (DIC=1059) or hyperbolic (DIC=1062) functions (Figure 1). The best fit model predicts that increasing density from 1 to 200 gametocytes µl-1 does not appreciably increase infectivity.”

*The uncertainty range is huge, and ranges from very little increase in infectivity with increasing gametocyte density, to very steep increases in infectivity with increasing gametocyte density. What steps do the authors propose are taken to reduce the uncertainty range*?

We agree that there is considerable uncertainty in the shape of the fitted curves reflecting the uncertainty in gametocyte density and the variability seen in mosquito feeding assays. We have included a few sentences outlining how the uncertainty can be minimised:

“The infectivity of mosquitoes to *Plasmodium*-infected blood is known to vary substantially within and between studies. This is in part due to methodological issues with the membrane feeding assay (5) but also related to biological differences in the parasite-vector combination and the blood environment (6). Further standardising the membrane feeding assay, improving the accuracy of QT-NASBA technique and the inclusion of additional covariates (such as estimates of gametocyte maturity which have been investigated in *P. vivax* (8)) would improve the accuracy of relationship between gametocyte density and mosquito infectivity.”

***2) Mosquito Infectivity***

*Equation A4 includes a parameter called kappa. This is confusing since kappa is conventionally used to represent the infectious reservoir*.

We have changed this parameter to eta to avoid confusion.

*Results: The results for age are presented as if there was evidence of an effect, but the confidence intervals suggest otherwise*.

Including age significantly improves the fit of the model though we appreciate that the 95% Bayesian credible intervals span 0. We have augmented the text in order to reflect this: “Including age improves the fit of the model suggesting age is an important confounder though the Bayesian credible intervals include zero.”

*The rationale for adjusting for asexual parasite density was not clear. It “improved model fit” but the aim here is not the best fitting model so much as to describe the relationship between estimated gametocyte density and infectivity. Including asexual parasite density as a main effect modifies the interpretation of the parameters*.

We appreciate that adjusting for asexual parasite density (as with age) may alter the shape of the relationship between gametocytes and mosquito infectivity, as asexual parasite density changes with increasing gametocyte density. However, the relationship between the density of gametocytes and asexual parasites is unlikely to remain constant (certainly not after certain drug treatments), so we feel that it is important to understand the impact of increasing gametocyte density without the influence of confounding variables. It also allows more accurate predictions of mosquito infectivity from cross-sectional surveys where these covariates are collected (such as in Burkina Faso). Since asexual parasite density and age are routinely collected we felt that the extra complexity of the model was warranted. We thank the reviewers for bringing this to our attention and we have changed the end of the Introduction accordingly to stress the rationale for doing so: “Other covariates that may influence infectivity such as asexual parasite density (measured by microscopy) and host age were also included *to increase the ability of the model to predict human to mosquito transmission*.”

***3) The Paper Estimates the Relationship Between Gametocyte Densities and Age***

*The data comes from a cross-sectional survey of 412 people in a village in Burkina Faso. There is likely to be seasonal variation in parasite densities and the relative densities in adults and children may vary seasonally due to different levels of acquired immunity. It should be stated when the survey was carried out and also that a single survey is a potential limitation*.

Thank you for highlighting this oversight. We have included a sentence to highlight the dangers of cross-sectional surveys in the main text. In the Results: “In the crosssectional survey in Burkina Faso children harbour more gametocytes than adults, with 10-year olds having 5 times as many gametocytes compared to 50-year olds (Figure 2).” In the Discussion: “The reservoir of infection is likely to vary between endemicity settings and according to local malaria control practices. It is also likely to vary over time so multiple cross-sectional surveys may be required.”

***4) Efficacy Definition***

*A major issue is to be quite clear what is meant by efficacy throughout the paper. This lack of clarity permeates this entire technical area (i.e., not only this manuscript) and causes a lot of confusion. I believe most or all instances of use of the word efficacy in this manuscript refers to a % reduction in the gametocyte density. How does this relate to % reduction in proportion of infected mosquitoes and how will this translate to reduced incidence of human infection? This is a critical discussion that is omitted. Discussion of efficacy without some idea what this translates to in terms of reduction in human infection is rather misleading*.

*If the authors mean reduced transmission from humans to mosquito, I suggest they use a narrative phrase, such as this and do not use the word efficacy at all*.

We appreciate the ambiguity of the term “efficacy” in the literature and we thank the reviewers for highlighting it here. As suggested we have changed the terminology so that each time the word efficacy is used we clarify which life-stage is targeted using a narrative phrase. We have also included the following sentences in the Discussion, stating:

“Figure 1 can be used to estimate how a reduction in the number of gametocytes will equate to a reduction in the proportion of mosquitoes becoming infected, and hence mosquito to human transmission. How this relates to the incidence of malaria and subsequent disease will depend, in part, on the degree of immunity in the human population.”

***5) Data/Model Description***

*The data the model was fitted to was not well described (“repeatedly testing samples with known gametocyte density”). I also had trouble working out whether the fitting was simultaneous with the model for mosquito infectivity or not. The text mentions that the “estimates were used in the functions below” but the table presents all parameters together. My concern is that that the error in the estimated gametocyte densities, from the QT-NASBA results and also the chance of gametocytes being contained in the blood sample, is incorporated*.

The uncertainty in the QT-NASBA technique includes the variability in the number of gametocytes in the blood sample. We have added a number of sentences stressing this and that the model for quantifying gametocyte density was fit at the same time as the model determining mosquito infectivity. This allows the uncertainty in gametocyte density to be reflected in the uncertainty of the best-fit line (and other covariates).

We have changed the “Mosquito infectivity” section of the methods:

“Gametocyte density was quantified using QT-NASBA and the uncertainty in the density estimates (the intra-assay variability) was quantified by fitting a hierarchical model to the 16 independent dilution series. In the Burkina dataset multiple assays were carried out on the same blood sample (an average of 2.63 assays per unknown blood sample). The mean gametocyte density from these multiple assays was taken to increase the accuracy of the estimates. The precise shape of the relationship was determined by fitting a range of different functional forms (a modified constant, linear, power, hyperbolic, sigmoid and Gompertz functions) and statistically determining which gave the best fit (a full description of the model is given in the Supplementary files). The model quantifying the uncertainty in gametocyte density estimates was fitted at the same time as the regression determining the relationship between gametocyte density (and other covariates) and the proportion of mosquitoes infected using Bayesian Markov Chain Monte Carlo methods. Fitting the models simultaneously enables the uncertainty in the gametocyte density estimates to be reflected in the uncertainty of the best fit model.”

*The model and parameters are not explained. The methods section merely mentions “a mathematical model” and the supplementary file gives an equation involving three parameters, but it is not clear how the equation was derived or what the parameters represent. A smooth curve is produced, but the relationship does not necessarily have to be smooth: for example, pregnant women may have higher parasitaemia and affect the age-curves*.

We have explained the age profile model in greater detail in the main text. We appreciate that certain groups, such as pregnant women, may have higher parasite densities, though we do not have more detailed information from the cross-sectional survey to include additional covariates within the model. Nevertheless, we feel that looking at things at a population level using smoothed curves is important for policy makers. The age profile takes into account how these heterogeneities, such as pregnancy status, changes with age and will result in elevated parasite levels in particular age groups.

The following section was included in the “Age profile of gametocyte density” section of the methods:

“A three parameter functional form was used which can describe an age profile which increases with age or peaks at an intermediate age. As above, a hierarchical model was used to estimate the gametocyte density and its associated uncertainty at the same time as the age profile allowing the full uncertainty of the shape of the function to be expressed (a full description of the data and model is given in the Supplementary files).”

*Figure 2 suggests that the relationship does not fit well for two of the age groups. This should be acknowledged and possible reasons mentioned*.

We agree that these two data points could be closer to the line, though the confidence intervals of the point estimate and the model do overlap. The functional form is flexible enough to pass much closer to these points though doing so generates a worse fit to these data. We feel that the main reason for the apparent discrepancy is the sampling from the highly overdispersed data, where missing the rare very high parasite densities by chance would result in lower mean estimates. Neither of these data points have particularly large sample sizes, which increases the probability of this happening. This overdispersion is taken into account in the fitting process and though other methods of central tendency could be presented in the figure (such as the geometric mean or median), since the model is showing the arithmetic mean we felt this was the best comparison. To highlight this we have added the following sentence to the legend of Figure 2: “In Figure 1 the [10-15) and [15-20) groups appear lower than the best fit line due to sampling artefacts generated by the highly overdispersed data.”

*A negative binomial error structure was used. The [Supplementary-material SD3-data] table suggests that there were some zeros in the data and that the densities are skewed. Did the residuals suggest a good fit? If there are many zeros, then a zero-inflated negative binomial model may be more appropriate*.

Though there are a lot of zero counts, this is adequately described by the simple negative binomial distribution. A zero-inflated negative binomial distribution was fitted to the estimates of gametocyte density (used to generate the gametocyte age profile) using maximum likelihood, though the improved fit didn’t warrant the extra complexity (likelihood ratio test p value = 0.7). Diagnostics on the residuals indicate no major issues other than the high level of uncertainty. We have changed the following section of the Discussion to reflect this: “However, the highly overdispersed distribution of gametocytes between hosts (adequately described by the negative binomial distribution) means that in the population from Burkina Faso…”

***6) Acknowledge Limitations***

*On a couple of occasions, the authors make statements that are not backed up by evidence from this paper*.

We have removed the misleading statements such as those highlighted below.

*Discussion. It is implied that this work gives direct evidence of the effectiveness of parasite strategies at low gametocyte densities such altering the sex ratio, but no direct information on strategies is obtained*.

We were trying to suggest that transmission at very low gametocyte densities is evidence that the strategies of the parasite to maximise transmission were working. However we agree that this sentence could be misleading and so we have deleted it from the Discussion.

*Although the paper does include an elegant sampling approach to quantify uncertainties, there should be a beefed up discussion of the uncertainties underlying this work. The authors should provide more discussion on what is known about which parameters in the input data are driving uncertainty, how much confidence they have that the model they chose is superior to alternative models, and on what basis they made this decision. There should also be a discussion on the limitations related to the differences between mosquitoes used in feeding assays, and wild-type mosquitoes*.

We agree that the uncertainties and the difference between laboratory and wild mosquitoes should be stressed further. We have added the following section to the Discussion:

“Gametocyte density measurement error is likely to cause the majority of the uncertainty seen in model outputs though the complexity of the membrane feeding assay is also likely to contribute. The infectivity of mosquitoes to Plasmodium infected blood is known to vary substantially within and between studies. This is in part due to methodological issues with the membrane feeding assay (5) but also related to biological differences in the parasite-vector combination and the blood environment (6). Further standardising the membrane feeding assay, improving the accuracy of QT-NASBA technique and the inclusion of additional covariates (such as estimates of gametocyte maturity which have been investigated in P. vivax (8)) would improve the accuracy of the relationship between gametocyte density and mosquito infectivity. It would also permit to see if the patterns described here are consistent across time and space, and enable a wider range of functional forms to be tested. Care should also be taken when interpreting the results of feeding assays as the mosquitoes used and their biting behaviour is likely to be different from that seen in wild mosquitoes in field situations (5).”

*Although uncertainty is presented in one of the figure panels, and this is very helpful, uncertainty ranges/confidence intervals should be quoted in the text where figures are provided*.

We agree and have added 95% Bayesian Credible Intervals where appropriate.

*We suggest that the authors consider how the paper can act as an aid to conceptualising the components underlying person-to-person transmission in field settings, and to gaining a better understanding of the uncertainties related to the component of transmission that their results speak to, and also to gain a better understanding of what the existing data gaps are that must be filled in order that the research community can determine how to test new transmission-reducing interventions, and quantify their effect on person-to-person transmission*.

We very much agree with the reviewers that this is an important topic. Person-to-person transmission is highly complex due to the complexities of the human immune response. Since this is not the topic of this manuscript, we concentrate our answers on the reservoir of infection (i.e., transmission, not necessarily incidence of malaria). We have augmented the following section of the Discussion to reflect our opinions on the subject:

“The contribution of different age groups to overall transmission will depend on local demography, age-dependent protection by malaria interventions (such as the use of bednets or time spent in the house and therefore personally protected by indoor residual spraying), and human to mosquito contact patterns. The last two of these are difficult to measure and poorly understood, reducing our ability to predict accurately the sources of infection. The reservoir of infection is likely to vary between endemicity settings and over time so multiple cross-sectional surveys may be required.”